# ACSL4 Directs Intramuscular Adipogenesis and Fatty Acid Composition in Pigs

**DOI:** 10.3390/ani12010119

**Published:** 2022-01-04

**Authors:** Hongyan Ren, Haoyuan Zhang, Zaidong Hua, Zhe Zhu, Jiashu Tao, Hongwei Xiao, Liping Zhang, Yanzhen Bi, Heng Wang

**Affiliations:** 1Key Laboratory of Animal Embryo Engineering and Molecular Breeding of Hubei Province, Institute of Animal Science and Veterinary Medicine, Hubei Academy of Agricultural Sciences, Wuhan 430064, China; renhy@hbaas.com (H.R.); zaidonghua@hbaas.com (Z.H.); zhuzhe@hbaas.com (Z.Z.); prof.hongwei.xiao@hbaas.com (H.X.); chzlp1982@hbaas.com (L.Z.); 2Key Laboratory of Agricultural Animal Genetics, Breeding, and Reproduction of the Ministry of Education, College of Animal Sciences and Technology, Huazhong Agricultural University, Wuhan 430070, China; hyzhang91@yahoo.com; 3Shandong Provincial Animal Husbandry General Station, Jinan 250022, China; muwhy@hotmail.com

**Keywords:** ACSL4, intramuscular, adipocyte, pig

## Abstract

**Simple Summary:**

In the livestock industry, intramuscular fat content is an important indicator of the meat quality of domestic animals. The variations of the Acyl-CoA Synthetase Long-Chain Family Member 4 (ACSL4) gene locus are associated with intramuscular fat content in different pig populations, but the detailed molecular function of ACSL4 in pig intramuscular adipogenesis remains obscure. Our study reveals the function of ACSL4 in pig intramuscular adipogenesis and provides new clues for improving the palatability of meat and enhancing the nutritional value of pork for human health.

**Abstract:**

The intramuscular fat is a major quality trait of meat, affecting sensory attributes such as flavor and texture. Several previous GWAS studies identified Acyl-CoA Synthetase Long Chain Family Member 4 (ACSL4) gene as the candidate gene to regulate intramuscular fat content in different pig populations, but the underlying molecular function of ACSL4 in adipogenesis within pig skeletal muscle is not fully investigated. In this study, we isolated porcine endogenous intramuscular adipocyte progenitors and performed ACSL4 loss- and gain-of-function experiments during adipogenic differentiation. Our data showed that ACSL4 is a positive regulator of adipogenesis in intramuscular fat cells isolated from pigs. More interestingly, the enhanced expression of ACSL4 in pig intramuscular adipocytes could increase the cellular content of monounsaturated and polyunsaturated fatty acids, such as gamma-L eicosapentaenoic acid (EPA), docosahexaenoic acid (DHA). The above results not only confirmed the function of ACSL4 in pig intramuscular adipogenesis and meat quality attributes, but also provided new clues for the improvement of the nutritional value of pork for human health.

## 1. Introduction

Intramuscular fat (IMF) content is an integral part of meat quality and directly influences meat tenderness, juiciness, and flavor. IMF refers to the chemically extractable fat inside the muscle, predominantly from intramuscular adipocytes, which are derived from preadipocytes that reside in the muscle [1]. However, the underlying mechanisms controlling the adipogenic differentiation and fat deposition of porcine intramuscular preadipocytes remain poorly understood, and obviously involve genetic, nutritional, and environmental factors [2].

The IMF content varies in different pig breeds and even in different individuals within the same breed populations. Dozens of functional or candidate genes have been identified and genetic polymorphisms associated with IMF have also been revealed [3]. Among them, the ACSL4 is one of the most frequently identified candidate genes related to IMF content in different population-based association studies in pigs [4,5,6,7,8]. The genomic variations around the ACSL4 locus control the ACSL4 transcription, but whether the fluctuations of ACSL4 expression levels could impact the intramuscular fat deposition remains unclear. A previous study reported that ACSL4 was involved in preadipocyte differentiation in pigs [9]. However, detailed functional validation of pig ACSL4 in the intramuscular adipogenesis was not thoroughly investigated.

The ACSL4 gene encodes fatty acid-CoA ligase 4, an isozyme of the long-chain fatty-acid-coenzyme ligase family. It converts free long-chain fatty acids into fatty acyl-CoA esters, and thereby plays a key role in lipid biosynthesis and fatty acid degradation [10]. Five isoforms of ACSL have been identified in humans and rodents, and they perform individual functions in fatty acid metabolism. ACSL4 showed a marked preference for arachidonic and eicosapentaenoic acid as substrates and play important roles in metabolic regulation of cell proliferation, differentiation and migration [11]. Dysregulation of ACSL4 has been demonstrated to promote ferroptosis in different cell types [12,13], and cause diverse diseases and disorders, such as hepatocellular carcinoma [14], breast cancer [15], intellectual disability [16], and mental dysfunction [17]. However, the expression dynamics and regulatory function of ACSL4 in adipogenesis or lipogenesis have not been thoroughly investigated.

In the present study, we explored the expression pattern and regulatory roles of ACSL4 gene during the differentiation of pig intramuscular pre-adipocytes and identified pig ACSL4 as a positive regulator of intramuscular adipogenesis. Further biochemical analysis revealed that ACSL4 could modulate the fatty acid composition in the pig intramuscular adipocytes and thus improve the nutritional value of meat with enriched polyunsaturated fatty acids.

## 2. Materials and Methods

### 2.1. Animals and Tissues

The animal experiments were approved by the Animal Care Committee of the Institute of Animal Science and Veterinary Medicine, Hubei Academy of Agricultural Sciences. The neonatal Yorkshire pigs were taken from the affiliated farm of the Institute of Animal Husbandry and Veterinary Medicine, Hubei Academy of Agricultural Sciences. The tissues (heart, liver, spleen, lung, kidney, longissimus dorsi, and adipose tissue) were collected from three 3 month-old castrated boars and stored at liquid nitrogen.

### 2.2. Primary Cell Isolation, Proliferation and Differentiation

Pig longissimus dorsi muscle tissue was cut with scissors into approximately 1 mm sections under sterile conditions and digested with collagenase type II for 2 h at 37 °C in a shaking water bath. The digested tissue was first centrifuged at 100× *g* for 1 min, and the resulting floating cells were collected in DMEM at 37 °C. The non-adherent cells were mostly intramuscular fat precursor cells. After adding fresh complete culture medium (89% DMEM/F12 medium + 10% FBS + 1% penicillin), most of the cells adhered to the culture dish after 2 days. When the cell confluence reached 90%, trypsinization was used to digest the cells and the medium was changed every 2 days.

For the proliferation assay, pig preadipocytes were seeded in 96 well plates at 4 × 10^3^ cells per well with the complete medium. CCK-8 solution was added to each well and incubated at 37 °C, 5% CO_2_ for 1.5 h, and the 450 nm absorbance was measured with a microplate reader (Meigu Molecule, Shanghai, China). The linear growth graph was drawn according to the OD value. For the cell differentiation analysis, the intramuscular adipocytes were cultured to complete confluence for 2 days, growth medium was replaced with differentiation medium (89% DMEM/F12 medium + 10% FBS + 1% penicillin + 5 µg/mL insulin + 0.5 µmol/L 3-isobutyl-1-methylxanthine + 1 µmol/L dexamethasone) for at least 3 days and lipid droplet should be visible.

### 2.3. Oil Red O Staining

The differentiated adipocytes were washed three times with PBS and fixed in 4% paraformaldehyde for 30 min, followed by washed with deionized water and incubated with 60% Oil Red O solution for 10 min. The cells were protected from light and washed with PBS three times. The images were captured using a microscope (Leica 4000B, Wetzlar, Germany). Oil Red O quantification was performed by measuring the positive area in three different fields (stained pixels were measured using Image J).

### 2.4. siRNA Knockdown

In our study, the ACSL4 siRNAs were designed according to their mRNA sequence (GenBank NM001038694.1) with the free online design tools (https://www.thermofiser.com/us/en/home/brands/invitrogen/ambion.html, accessed on 21 February 2020), and all the siRNA sequences were synthesized by Shanghai Genepharm Co.,LTD. The sequences are as follows. ACSL4-siRNA: 5′-GTCCAAGAGATGAATTATATT-3′; a scramble sequence 5′-GTTCTCCGAACGTGTCACGT-3′ was also synthesized as a negative control. Transfection was conducted with Lipofectamine 3000 (Invitrogen, Carlsbad, CA, USA) according to the manufacturer’s protocol. The preadipocytes were transfected with 5 µL Lipofectamine 3000 with 100 pmol siRNA-ACSL4 or siRNA-NC in 125 µL Opti-MEM™ media and the fluorescence of FAM group was observed. After 72 h, the cells were harvested for further analysis.

### 2.5. Adenovirus Transduction of Intramuscular Preadipocytes

For overexpression analysis, the ACSL4 CDS sequence was cloned into the Y4261 adenovirus plasmid and transfected in HEK293 cells to package the recombinant adenoviral vector containing the ACSL4 gene, according to the manufacturer instructions (Genepharm Co., Ltd., Shanghai, China). The pig intramuscular preadipocyte cells were seeded into 12 well culture plates at 80% confluence and the adenovirus was added to the medium. After 48 h of transfection, the cells were collected and total RNA and protein were extracted for expression analysis. Adenovirus ADV4-ACSL4 and ADV4-NC (both at 1011 pfu/mL) were synthesized by Gene Pharma. The preadipocytes were cultured in a six-well plate, and adenovirus was added (MOI = 50) when the cells reached 80–90% confluence. Fluorescence expression was observed after 48 h. The cells were collected 72 h later for further analysis.

### 2.6. RNA Extractions, cDNA Synthesis and Realtime PCR

Total RNA from porcine tissues and cells was isolated using a total RNA extraction kit and a reverse transcription kit (TIANGEN, Beijing, China), according to the manufacturer’s instructions. The qPCR primers of ACSL4, FASN, ACACβ, and C/EBPα were designed according their cDNA sequence (GenBank NM_001033600.1; NM_007988.3; XM_006530111.4; NM_001287514.1) with free online design tools (https://www.ncbi.nlm.nih.gov/tools/primer-blast/index.cgi, accessed on 18 March 2020) The details of the realtime-PCR primer sequences is in Appendix A and the relative quantification of (m)RNA expression was analyzed according to standardized GAPDH content adopting the method of 2^−ΔΔCT^.

### 2.7. Western-Blot Analysis

The total cellular protein extracts were collected in the RIPA Lysis and Extraction Buffer (Thermo Fisher, Waltham, MA, USA). Protease inhibitor (PMSF) at a concentration of 1% was added into the RIPA buffer to avoid proteolysis. For each sample, 50 μL protein lysate collected in a 1.5 mL centrifuge tube was treated in an ice bath for 30 min, followed by centrifugation at 10,000× *g* at 4 °C for 15 min. Protein supernatant in the tube was transferred into a new tube and stored at 4 °C for further protein quantification by BCA protein assay kit (P0006, Beyotime, Shanghai, China). In all, 20 μg of protein extract for each sample was subjected to 12% sodium dodecyl sulfate-polyacrylamide gel electrophoresis (SDS-PAGE Gel Preparation Kit, Beyotime) and transferred to polyvinylidene difluoride (PVDF) membrane (Bio-rad, Berkeley, CA, USA). The membranes were blocked in a TBST (150 mM NaCl, 20 mM Tris-HCl at pH 8.0, 0.05% Tween 20) blocking buffer with 5% non-fat dry milk powder at room temperature for 1 h. After three washes with the TBST buffer, the membranes were then incubated with primary antibodies overnight at 4 °C with shaking. After three washes in the TBST buffer, blots were incubated with secondary antibodies. The protein bands were then visualized by ECL detection reagent (Beyotime). The band identification and quantification were conducted using a ChemiDoc™ XRS+ System and Image Lab Software (BioRad). The primary antibodies used for Western blotting were rabbit monoclonal antibody against the mouse ACSL4/FACL4 (ab205199, Abcam; dilution 1:1000) and mouse monoclonal antibody against the human GAPDH (60004-1-Ig, proteintech, Wuhan, China; dilution 1:2000). The HRP-conjugated secondary antibodies were anti-mouse and anti-rabbit IgG (7076 and 7074, respectively, Cell Signaling Technology; dilution 1:2000). The intensity of the protein bands was measured by the Image Lab software package.

### 2.8. Transcriptome Analysis by RNAseq

After a 48 h transfection of the adipose precursor cells with siRNA, the samples were lysed with TRIzol reagent and stored in a refrigerator at −80 °C. The RNA-seq library for each sample was constructed by Beijing Compass Bio-Technology Co., Ltd., Beijing, China (www.kangpusen.com, accessed on 20 June 2020) based on the protocols of Illumina HiSeqTM2500/MiSeq™ to generate paired-end reads. The quality of the RNA-seq reads from all the samples was checked using FastQC (0.11.5, Babraham institute, Cambridge, UK). The reads that passed the quality control were mapped to the sus scrofa genome from Ensembl using STAR program (2.5.2a).

### 2.9. Fatty Acid Analysis by Gas Chromatography-Mass Spectrometry

The adipose precursor cells were transfected with adenovirus vector. 1 × 10^7^ cells were harvested for each cell sample and resuspended with 1 mL of chloroform-methanol (2:1) solution. The samples were snap frozen in liquid nitrogen and then dissolved at room temperature. The samples were ground at 60 Hz for 2 min, followed by ultrasonic treatment for 30 min. The samples were then centrifuged at 13,000 rpm for 5 min at 4 °C and the supernatant was collected and mixed with 2 mL of 1% methanolic sulphate. The mixed samples were esterified at 80 °C for 30 min and 1 mL of hexane was added for extraction. The samples were washed at 4 °C with 5 mL dd H_2_O, followed by sodium sulphate powder anhydrous to remove extra water. The supernatant was collected by centrifuging at 13,000× *g*rpm for 5 min and mixed with 25 μL of methyl salicylate. The supernatant from the mixture was placed on a DB5 capillary column (30 m × 0.25 mm) for gas chromatography mass spectrometry (Agilent 6890N/5975B) analysis.

### 2.10. Statistical Analysis

The numbers of biological replicates and technical repeats in each experimental group were three or more. The statistical analyses were performed using Graphpad Prism (Graphpad Software 6.0, Chicago, IL, USA). The data are expressed as means + SEM or SD. *p*-value of 0.05 is considered statistically significant.

## 3. Results

### 3.1. The Isolation, Proliferation, and Differentiation of Porcine Intramuscular Preadipocytes

The differential velocity adherent technique was employed to isolate the intramuscular preadipocytes from the pig muscles. After their attachment to the surface of the culture dishes, the cells stretched out like fibroblasts in morphology and staggered protrusions between adjacent cells were easily observed (Figure 1A). In the growth media, the intramuscular fat precursor cells showed an S-shaped growth curve (Figure 1B). The cells proliferated robustly during the first 2 to 3 days of inoculation, and reached the plateau phase after 4 days. In order to examine the fat deposition and lipid droplet morphology in the cultured intramuscular adipocytes, the cell differentiation was induced and stained with Oil Red O. After 3 days of induction, a small amount of lipid droplets was detected (Figure 1C). The abundance of lipid droplets gradually increased from day 3 to day 9, and a large number of lipid droplets appeared on day 9 (Figure 1C). Therefore, we successfully established the intramuscular adipogenesis system with porcine intramuscular pre-adipocytes for subsequent molecular and cellular functional studies.

### 3.2. The Spatial-Temporal Expression Pattern of Pig ACSL4 Gene

We first examined the expression of ACSL4 gene in different tissues from the developing animals which are depositing intramuscular fat aggressively. We found relatively high expression of ACSL4 in the liver, lung and spleen (Figure 2A), suggesting that the liver is the major organ for fatty acid synthesis. However, the ACSL4 expression in the muscle tissues was much lower compared to other organs, indicating that ACSL4 is only expressed in specific type of cells within the bulk muscle tissue, or it is not the predominant fatty acid-CoA ligase isoform in the developing muscle. Therefore, we further validated the ACSL4 expression in the purified intramuscular preadipocytes during the adipogenic differentiation program. We detected both the mRNA and protein expression of ACSL4 in the preadipocytes purified from the pig skeletal muscle. The mRNA and protein levels of ACSL4 gradually increased and peaked on day 3 during the adipogenic differentiation process (Figure 2B,C and Appendix A), when most of preadipocytes differentiated into adipocytes. Subsequently, the expression of ACSL4 showed a steady declining trend, suggesting that ACSL4 was induced in the early stage of adipogenic differentiation and remained low during adipocyte maturation.

### 3.3. Knockdown of ACSL4 in Intramuscular Preadipocytes Impairs Fat Deposition and Cell Development

Next, we wondered whether the knockdown of ACSL4 gene in intramuscular preadipocytes cells affects the gene expression associated with fat deposition. We performed RNA-seq to measure the expression profiles of intramuscular preadipocyte cells transfected with siRNA-NC and siRNA-ACSL4. The RT-qPCR and Western blot showed that the knockdown efficiency could reach more than 70% and 80% for mRNA and protein levels, respectively (Figure 3A,B and Appendix A). Next, we performed transcriptome analysis in the ACSL4 knockdown cells and the control cells to discover the differentially expressed genes (DEG) and related pathways caused by the ACSL4 loss-of-function. A total of six RNA-seq libraries were constructed. The sequencing reads data and all the expressed genes are summarized in the Appendix A (Appendix A). Compared to the control, 134 genes were differentially expressed in the ACSL4 knockdown cells (Fold Change ≥ 2, *p* < 0.05). Among the DEGs, we identified that many adipogenesis-related genes were downregulated, including MOGAT1 [18], IL10 [19], TFAP2B [20], and ACSL6 [21], while several negative regulators of adipogenesis, such as FABP4 [22] and SST [23], were upregulated (Figure 3C). The KEGG pathway analysis of the DEGs shows that the majority of differentially expressed genes were enriched in signaling pathways that are important during adipogenesis, such as “response to ATP” and “cellular response to growth factor stimulus” [24]. Interestingly, several cellular development and differentiation pathways, such as “cell exploration behavior” and “multicellular organismal response to stress” were also enriched in the KEGG analysis of the DEGs (Figure 3D), indicating that ACSL4 could be involved in basic cell development and differentiation processes. These results indicate that ACSL4 is necessary for the normal adipocyte differentiation and could be responsible for lipid composition of cell membrane by fatty acid oxidation or lipid production.

### 3.4. Overexpression of ACSL4 Results in Enhanced Lipid Deposition and Elevated Polyunsaturated Fatty Acids Synthesis in Pig Intramuscular Adipocyte

To further illustrate the function of ACSL4 in intramuscular adipogenesis and lipogenesis, we performed an adenovirus-mediated overexpression of ACSL4 and evaluated the lipid deposition and composition in pig intramuscular preadipocytes. Compared to the control, the transduction of the cells with Adeno-ACSL4 dramatically increased the ACSL4 protein expression and lipid deposition, as shown by the Western blot (Figure 4A and Appendix A) and Oil Red O staining (Figure 4B). We also observed significantly elevated adipogenic marker gene expression including FASN, ACACB, and C/EBPα (Figure 4C). Thus, it is concluded that ACSL4 is a positive regulator of intramuscular adipogenesis in pigs.

Next, to understand how ACSL4 was involved in the de novo lipogenesis in pig muscle, we applied mass spectrometry-based lipidomics to determine how ACSL4 overexpression affects the composition and distribution of fatty acids in intramuscular preadipocytes. The quantification of the lipid classes revealed dramatic ACSL4-induced changes in the abundance of lipid species in several of the analyzed lipid classes. The overall abundance of saturated fatty acids (SFAs), which are the most abundant lipid classes in intramuscular adipocytes, was not affected by ACSL4 overexpression (Table 1). By contrast, we detected an approximately 22% and 29% increase in the concentration of mono- and polyunsaturated fatty acids (MUFA and PUFA, respectively) in response to ACSL4 overexpression (Table 1). Further analysis showed that this increase was caused by increased levels of MUFA containing C16:1, C18:1, and C24:1 and PUFA containing C18:2, C18:3N6, and C20:4N6 (Table 1). Both the n-3 fatty acid and n-6 fatty acid increased after ACSL4 overexpression but the n-6/n-3 ratio remained stable.

## 4. Discussion

The quality of pork is closely related to the IMF content, so understanding the molecular mechanisms of intramuscular adipogenesis is important for improving pork quality. Numerous studies indicated that the role of the ACSL4 gene is associated with the IMF content in different pig populations, but the intrinsic biological function of ACSL4 in porcine intramuscular adipogenesis is still unclear. To explore the possible roles of ACSL4 in intramuscular adipogenesis, we first established and optimized the adipogenic differentiation procedure by using the porcine preadipocytes derived from skeletal muscle. We showed that pig ACSL4 expression levels increased during adipogenic induction and decreased to the levels comparable to the pre-differentiation stage. The loss-of-function experiments by siRNA knockdown in preadipocytes showed that ACSL4 is necessary for the full proceeding of the early adipogenic differentiation transcription program. It suggests that the ACSL4 could stimulate adipogenesis and is more effective in the early stage of adipogenic differentiation. Therefore, consistent with other studies, ACSL4 is a positive regulator of adipogenic differentiation of adipocytes [9,25,26], epithelial cells [27], and smooth muscle cells [28] from different tissue origins. Our data also showed that ACSL4 expression remains low in the matured adipocytes, indicating that ACSL4 is dispensable for the maintenance of fat cells and lipid storage. However, the ACSL4 gene could also inhibit adipogenic differentiation in cells other than adipocytes, such as bone marrow-derived macrophages [29]. Thus, it is possible that ACSL4 may produce distinct effects in different tissues and thus the function of ACSL4 in adipogenesis is context-dependent. We also noted the high expression profile of ACSL4 in lung tissue. Several studies have revealed that ACSL4 is mainly expressed in peroxisomes and the ER, while the process of lung lipid metabolism in lung tissue requires the involvement of substantial numbers of peroxisomes [30,31]. We postulate that the critical effect of peroxisomes in lung tissue causes the hyper-expression profile of ACSL4 in lung tissue.

Based on the effects of ACSL4 interference on the preadipocyte transcriptome, we chose to focus our analyses of the lipogenesis and lipidome on fatty acid compositions. Our results clearly showed that ACSL4 overexpression can significantly increase the adipogenesis and lipid accumulation in porcine intramuscular preadipocytes, as demonstrated by the Oil Red O staining and marker gene expression. Since the higher ratio of polyunsaturated (PUFA) to saturated fatty acids (SFA) and a more favorable balance between n-6 and n-3 PUFA is preferred in meat products [32], we examined the fatty acid composition in ACSL4-overexpressed porcine intramuscular preadipocytes. Mass-spectrometry based lipidomics revealed the extensive remodeling of lipid compositions in response to ACSL4 overexpression. Of the major lipid classes analyzed, ACSL4 overexpression led to the enrichment of several species of MUFA and PUFA. Among them, γ-linolenic acid, eicosapentaenoic acid (EPA), docosahexaenoic acid (DHA), and arachidonic acid increased by 50%, 54%, 32%, and 25% respectively. The EPA and DHA of n-3 PUFA are potent lipid mediators that are incorporated in many parts of the body and are essential in various biological processes, such as brain development, as well as anti-inflammatory and anti-aging processes [33]. PUFAs play an important role throughout life, but our human bodies do not efficiently produce n-3 fatty acids such as EPA and DHA. Therefore, EPA and DHA are essential in the human diet; they are mainly provided by marine sources, such as fish oil supplements [34]. PUFAs are often present at very low levels in the meat of domestic animals, especially those of the n-3 series, which have particularly beneficial effects on health. Regarding the dramatic differences in oleic acid, we found that ACSL4 is an essential contributor for triggering cellular ferroptosis [12,35]. Meanwhile, a previous study showed that oleic acid reduced the mortality of ferric death cells [13]. We conclude that differences in intracellular oleic acid may respond antagonistically to ACSL4 overexpression. Animal scientists are consistently searching for new ways to change meat fatty acid composition, mainly through feeding plant or fish oil sources of PUFA to animals [36]. The current study provided new possibilities to increase PUFA levels through the genetic selection or manipulation of the ACSL4 gene in pigs to enhance PUFA production and the nutritional value of pork products.

## 5. Conclusions

In summary, our loss-of-function and gain-of-function experiments confirmed that ACSL4 is directly involved in pig intramuscular adipogenesis. Moreover, the lipidomics analysis also indicated that ACSL4 could be a potential target for the improvement of the nutritional value of pork for human health.

## Figures and Tables

**Figure 1 animals-12-00119-f001:**
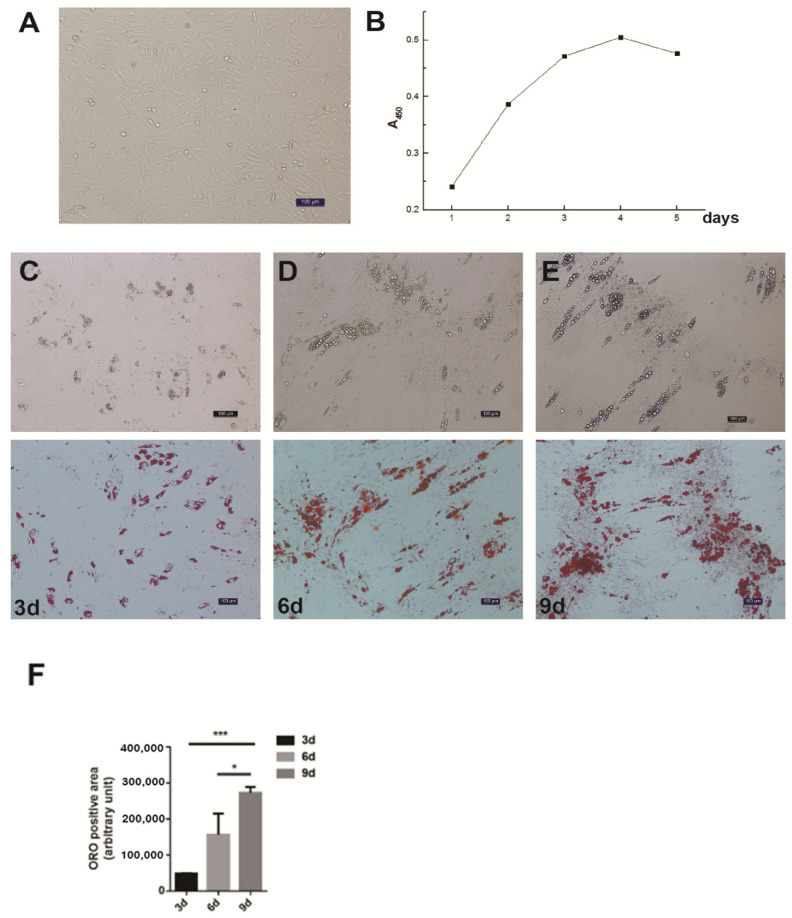
Isolation, proliferation and differentiation of pig intramuscular preadipocytes. (**A**) Purified intramuscular preadipocytes from pig skeletal muscle. (**B**) The proliferation dynamics of porcine intramuscular preadipocytes in culture. (**C**–**E**) The adipogenic differentiation of porcine intramuscular preadipocytes at 3, 6, and 9 days. The upper panels are bright view and the lower panels are Oil Red O staining. (Scale bars, 50 µm). (**F**) The quantitation of Oil Red O (ORO)-positive region for porcine intramuscular preadipocytes at 3, 6, and 9 days. Data are expressed as means + SEM. * indicated *p* < 0.05, *** indicated *p* < 0.001.

**Figure 2 animals-12-00119-f002:**
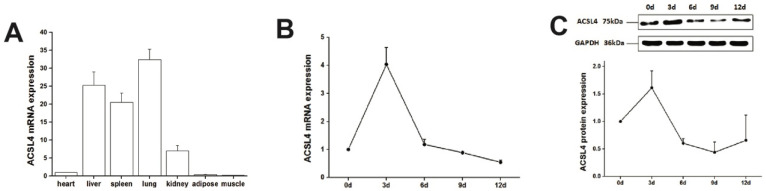
The pig ACSL4 expression pattern in different tissues and during adipogenic differentiation. (**A**) The ACSL4 gene expression in different pig tissues including heart, liver, spleen, lung, kidney, fat, and skeletal muscle. (**B**) The mRNA expression levels of pig ACSL4 during adipogenic differentiation of intramuscular preadipocytes. (**C**) The protein levels of pig ACSL4 during adipogenic differentiation of intramuscular preadipocytes as shown by Western blot. Data are expressed as means + SEM.

**Figure 3 animals-12-00119-f003:**
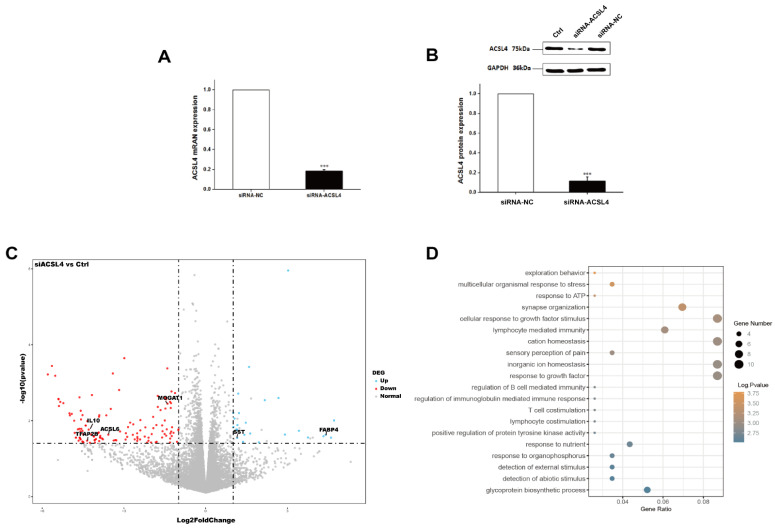
Transcriptome analysis of siRNA mediated ACSL4 knockdown in the pig intramuscular preadipocytes. (**A**) Realtime PCR showed ACSL4 gene was dramatically downregulated after siRNA knockdown. (**B**) Western blot indicated ACSL4 protein was dramatically reduced after siRNA knockdown compared to control. (**C**) Differentially expressed genes after ACSL4 knockdown as shown by the volcano plot. (**D**) KEGG pathway analysis of the differentially expressed genes after ACSL4 knockdown. Data are presented as the mean ± SEM. Comparisons were performed by unpaired two-tailed Student’s *t*-tests. *** indicated *p* < 0.001.

**Figure 4 animals-12-00119-f004:**
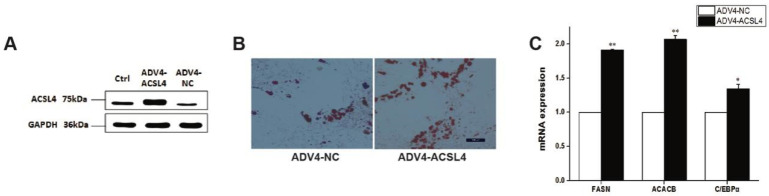
Overexpression of pig ACSL4 gene increased adipogenesis and lipid deposition in pig intramuscular preadipocytes. (**A**) Western blot showed that ACSL4 protein increased after adenovirus mediated overexpression. (**B**) Oil Red O staining showed that ACSL4 overexpression stimulated lipid deposition. (**C**) Realtime PCR showed that adipogenic genes FASN, ACACB, C/EBPa increased significantly after ACSL4 overexpression. Bars are presented as the mean ± SEM. Comparisons were performed by unpaired two-tailed Student’s *t*-tests. * indicated *p* < 0.05 and ** indicated *p* < 0.01.

**Table 1 animals-12-00119-t001:** The fatty acid composition of samples.

Fatty Acid	ADV4-ACSL4 (μg/1 × 10^7^ Cells)	ADV4-NC (μg/1 × 10^7^ Cells)
Saturated fatty acid	50.63	49.07
Monounsaturated fatty acid	24.38	19.98
Polyunsaturated fatty acid	37.87	29.27
n-3 fatty acid	19.35	14.55
n-6 fatty acid	15.00	12.023
n-6/n-3 fatty acid	0.78	0.83
C14:0 (Myristic acid)	0.83	0.63
C14:1 (Myristoleic acid)	0.22	0.23
C15:0 (Pentadecanoic acid)	0.80	0.62
C15:1 (Pentadecenoic acid)	0.14	0.14
C16:0 (Palmitic acid)	21.84	21.99
C16:1 (Palmitoleic acid)	1.69	1.30
C17:0 (Heptadecanoic acid)	1.30	1.04
C17:1 (Heptadecenoic acid)	0.58	0.41
C18:0 (Stearic acid)	24.54	23.61
C18:1 (Oleic acid)	16.63	13.73
C18:2 (Linoleic acid)	2.58	1.89
C18:3N6 (γ- linolenic acid)	0.24	0.16
C18:3N3 (Linolenic acid)	0.13	0.11
C20:0 (Arachidic acid)	0.69	0.64
C20:1 (Eicosenoic acid)	1.08	0.99
C20:2 (Eicosadienoic acid)	0.45	0.36
C20:3N6 (Eicosatrienoic acid triglyceride N6)	3.93	3.22
C20:3N3 (Eicosatrienoic acid triglyceride N3)	0.15	0.14
C20:4N6 (Arachidonic acid)	10.83	8.64
C20:5N3 (Eicosapentaenoic acid)	1.75	1.14
C21:0 (Heneicosanoic acid)	0.28	0.22
C22:0 (Behenic acid)	0.25	0.23
C22:2 (Docosadienoic acid)	0.49	0.45
C22:1N9 (Erucic Acid)	0.44	0.41
C22:6N3 (Docosahexaenoic Acid)	17.32	13.17
C24:0 (Lignoceric acid)	0.11	0.10
C24:1 (Nervonic acid)	3.59	2.79

Note: Cell sample is 1 × 10^7^ cells and was resuspended in 1 mL of chloroform-methanol solution. Units in the table means μg/1 × 10^7^ cells.

## Data Availability

All the sequencing data are available through the BioProject database, under the accession BioProjectID: PRJNA791163.

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
