# Peer review of "ACSL4 Directs Intramuscular Adipogenesis and Fatty Acid Composition in Pigs"

_animals, 2022, doi:10.3390/ani12010119_

Round 1
Reviewer 1 Report
In this manuscript, Hongyan et. al. aim to gain insight into the functional characterization of ACSL4 in the intramuscular adipogenesis in pig.
Minor comments:
Methods:
How many pigs where used and which was their sex?
The tense of verbs should be revised in the methods as they are mixed.
References showing that PCR primers work, or a description of how they were designed are missing.
The methods for WB are lacking details. In particular, primary antibody vendor and catalog numbers should be provided as well as the incubating conditions for each one.References to independent published studies or in-house data validating such antibodies in pig is strongly suggested.
How were siRNA sequences designed?
In general, the methods used for the investigation are lacking details. For all nucleic acid sequences and adenoviral experiments, the sequence design and controls should be explained.
Figure legend 1. Panels D and E have no reference in legend.
How and from where the tissues for Figure 2 panel A were harvested? “developing animals which are depositing intramuscular fat aggressively” these animals were not mentioned before.
Major comments:
It is really hard to follow the paper, as methods are not clearly explained in a logical progression. Authors should improve methods and present their results sequentially and in logical progression for the reader to follow.
Author Response
Response to Reviewer 1 Comments
Point 1: How many pigs where used and which was their sex?
Response 1: Thank you very much for this comment. We chose three 3-month-old castrated boars as experimental animals.
To take your suggestion into account we revised the method of the manuscript. Please find the following text added to the manuscript:
“The tissues (heart, liver, spleen, lung, kidney, longissimus dorsi and adipose tissue) were collected from three 3-month-old castrated boars and stored at liquid nitrogen.” (line 77-78)
Point 2: The tense of verbs should be revised in the methods as they are mixed.
Response 2: Thank you very much for your comment. We revised the tense of verbs in the methods according to your suggestion. please check the method section of the manuscript
“When the cell confluence reached 90%, trypsinization was used to digest the cells and the medium was changed every 2 days.” (line 86-87)
Point 3: References showing that PCR primers work, or a description of how they were designed are missing.
Response 3: Thank you very much for your comment. We described the primer design in detail in the methods section and provide the amplicon information in Table S1, please check the text in the manuscript.
“Total RNA from porcine tissues and cells was isolated using total RNA extraction kit and Reverse Transcription kit (TIANGEN, China) according to manufacturer’s instructions. The qPCR primer of ACSL4, FASN, ACACβ and C/EBPα were designed according their cDNA sequence (GenBank NM_001033600.1; NM_007988.3; XM_006530111.4; NM_001287514.1) with the free online designtools (https://www.ncbi.nlm.nih.gov/tools/primer-blast/index.cgi). The detail of realtime-PCR primer sequences was in TableS1 ” (line 129-137)
Table S1 Primers used for real-time quantitative PCR
|
Primer name |
Sequence (5′→3′) |
Length(nt) |
production length |
GC (%) |
Tm |
|
ACSL4-F |
GATTGACAGAATCGTGTGGCG |
21 |
187 |
52.38 |
59.94 |
|
ACSL4-R |
CCCATGGAGATATTCTGTCCACC |
22 |
52.17 |
60.24 |
|
|
FASN-F |
TCACCTACGAGGCCATTGTG |
20 |
327 |
55 |
59.75 |
|
FASN-R |
GCTTCAGCAGGACGTTG |
17 |
58.82 |
55.9 |
|
|
ACACβ-F |
CAGAACCTGCATGACGGAGT |
20 |
155 |
55 |
60.04 |
|
ACACβ-R |
TGAGGACAAACATGGTCGGG |
20 |
55 |
55.96 |
|
|
C/EBPα-F |
GGCAAAGCCAAGAAGTCGGTA |
21 |
150 |
52.38 |
60.88 |
|
C/EBPα-R |
ATTGTCACTGGTCAGCTCCA |
20 |
50 |
52.94 |
Point 4: The methods for WB are lacking details. In particular, primary antibody vendor and catalog numbers should be provided as well as the incubating conditions for each one. References to independent published studies or in-house data validating such antibodies in pig is strongly suggested.
Response 4: Thank you very much for your suggestion. We rewrote the western method in the manuscript, please find the following text in the manuscript
“Total cellular protein extracts were collected in the RIPA Lysis and Extraction Buffer (Thermo Fisher). Protease inhibitor (PMSF) at a concentration of 1% was added into the RIPA buffer to avoid proteolysis. For each sample, 50 μl protein lysate collected in a 1.5 ml centrifuge tube was treated with ice-bath for 30 min, followed by centrifugation at 10,000 × g at 4 °C for 15 min. Protein supernatant in the tube was transferred into a new tube and stored at 4 °C for further protein quantification by BCA Protein Assay Kit (P0006, Beyotime Institute of Biotechnology). In total, 20 μg of protein extract for each sample was subjected to 12% sodium dodecyl sulfate-polyacrylamide gel electrophoresis (SDS-PAGE Gel Preparation Kit, Beyotime) and transferred to polyvinylidene difluoride (PVDF) membrane (BioRad). The membranes were blocked in TBST (150 mM NaCl, 20 mM Tris-HCl at pH 8.0, 0.05% Tween 20) blocking buffer with 5% non-fat dry milk powder at room temperature for 1 h. After three washes with TBST buffer, the membranes were then incubated with primary antibodies, overnight at 4 °C with shaking. After three washes in TBST buffer, blots were incubated with secondary antibodies. Protein bands were then visualized by ECL detection reagent (BeyoECL Moon, Beyotime). Band identification and quantification were conducted using a ChemiDoc™ XRS+ System and Image Lab Software (BioRad). The primary antibodies used for Western blotting were rabbit monoclonal antibody against the mouse ACSL4/FACL4 (ab205199, Abcam; dilution 1:1,000), mouse monoclonal anti-body against the human GAPDH (60004-1-Ig, Proteintech, Wuhan, China; dilution 1:2,000). HRP-conjugated secondary antibodies were anti-mouse and anti-rabbit IgG (7076 and 7074, respectively, Cell Signaling Technology; dilution 1:2,000). The intensity of protein bands was measured by the Image Lab software package.” (line 139-160)
Point 5: How were siRNA sequences designed
Response 5: Thank you very much for your comment. We revised the siRNA method in the manuscript, please check the following text in the manuscript
“In our study, the ACSL4 siRNA were designed according to its mRNA sequence (GenBank NM001038694.1) with the free online design tools (https://www.thermofiser.com/us/en/home/brands/invitrogen/ambion.html), and all the siRNA sequences were synthesized by Shanghai Genepharm Co., LTD. The sequences are as follows: ACSL4-siRNA 5’-GTCCAAGAGATGAATTATATT-3’; a scramble sequence 5’-GTTCTCCGAACGTGTCACGT-3’ was also synthesized as a negative control. Transfection was conducted with Lipofectamine 3000 (Invitrogen, USA) according to manufacturer’s protocol.” (line 105-112)
Point 6: In general, the methods used for the investigation are lacking details. For all nucleic acid sequences and adenoviral experiments, the sequence design and controls should be explained.
Response 6:Thank you very much for your comment. We inserted the details of the adenovirus experiment into the manuscript, please review these revised texts in the manuscript.
“For overexpression analysis, the ACSL4 CDS sequence was cloned into the Y4261 adenovirus plasmid and transfected in HEK293 cells to package the recombinant adenoviral vector containing the ACSL4 gene according to the manufacturer instructions (Genepharm Co, LTD, Shanghai, China). Pig intramuscular preadipocytes cells were seeded into 12-well culture plates at 80% confluence and the adenovirus was added to the medium. After 48h of transfection, cells were collected and total RNA and protein were extracted for expression analysis. Adenovirus ADV4-ACSL4 and ADV4-NC (both at 1011 pfu/ml) were synthesized by Gene Pharma. Preadipocytes were cultured in a six-well plate, and adenovirus was added (MOI = 50) when cells reached 80-90% confluence. Fluorescence expression was observed after 48h. Cells were collected 72h later for further analysis.” (line 117-127)
Point 7: Figure legend 1. Panels D and E have no reference in legend.
Response 7: Thank you for your comment. We have added the panels in the figure legend. The text is as follows
“(C-E) The adipogenic differentiation of porcine intramuscular preadipocytes at 3, 6 and 9 days. The upper panels are bright view and the lower panels are oil red O staining (Scale bars, 50µm).” (line 119-202)
Point 8: How and from where the tissues for Figure 2 panel A were harvested? “developing animals which are depositing intramuscular fat aggressively” these animals were not mentioned before.
Response 8: Thank you for your suggestion. These developing animals were three 3-month-old castrated boars. We have further revised the method of the manuscript. Please check the following paragraphs in the manuscript
“The tissues (heart, liver, spleen, lung, kidney, longissimus dorsi and adipose tissue) were collected from three 3-month-old castrated boars and stored at liquid nitrogen.” (line 77-78)
Point 9: It is really hard to follow the paper, as methods are not clearly explained in a logical progression. Authors should improve methods and present their results sequentially and in logical progression for the reader to follow.
Response 9: Thank you for your suggestion. We reordered the methods section according to the logical progression, please check the methods section of the manuscript

Reviewer 2 Report
Extremely interesting work with great application potential. Innovative research that was well planned, carried out and described presents the function of ACSL4 in pig intramuscular adipogenesis and provided new clues for improving the palatability of meat and enhancing the nutritional value of pork for human health.
I recommend this article – “ACSL4 directs intramuscular adipogenesis and fatty acid composition in pig” to be printed in Journal Animals, Animal Genetics and Genomics, Transcriptomics, and Computational Biology for Biodiversity Studies and Quality-Related Traits Selection in Livestock.
Author Response
Reviewer2_Comment:
Extremely interesting work with great application potential. Innovative research that was well planned, carried out and described presents the function of ACSL4 in pig intramuscular adipogenesis and provided new clues for improving the palatability of meat and enhancing the nutritional value of pork for human health.
I recommend this article – “ACSL4 directs intramuscular adipogenesis and fatty acid composition in pig” to be printed in Journal Animals, Animal Genetics and Genomics, Transcriptomics, and Computational Biology for Biodiversity Studies and Quality-Related Traits Selection in Livestock.
Response: We appreciate your recognition and approval for this article.

Reviewer 3 Report
Thank you for providing the opportunity to review the manuscript entitled ‘ACSL4 directs intramuscular adipogenesis and fatty acid com-2 position in pig’. The authors have nicely developed intramuscular adipogenic cell culture, measured ACSL4 expression followed by loss and gain of function analyses. The authors have used adequate technical skills and expertise in addition to clear scientific reasoning for some sections. I have the following suggestions/comments.
Throughout the manuscript, the authors mention the genetic polymorphism of ACSL4 and its impacts but have not performed any experiment in relation to it. I would suggest rewriting so that the message gets through clearly according to the results of the manuscript.
Figure 2A- Have authors measured ACSL4 expression in intramuscular adipose tissue in addition to muscle and adipose? The authors explained liver-related high expression but did not comment on lungs (highest expression).
In Figure 1, the authors show a gradual increase in lipid accumulation from days 3 to 9 and claim that ACSL4 is positively associated with adipogenesis. However, ACSL4 expression increased during early adipogenesis and decreases at the maturation phase. The authors have not described or explained this difference.
In the ACSL4 knockdown experiment results, authors have mentioned only downregulated DEGs. They could mention upregulated DEGs that may reveal genes suppressing adipogenesis.
In general, the image quality should be increased. The authors could perform Oil Red O image analyses to measure the differences.
In Table 1, it is unclear why there is a break. Additionally, adding significance values could increase the validity of the differences and ease their identification. Moreover, a huge difference in Oleic Acid (most abundant FA in intramuscular fat) levels between overexpression and control samples is evident, authors haven’t discussed it.
Authors have performed RNA-Seq but I could not find if they have submitted the data to any scientific database portal e.g. NCBI etc.
Authors claim that their study is novel as studies investigating the role of ACSL4, mechanism of action and loss/gain of function explorations were lacking. However, a study has previously been published demonstrating the miRNA regulation of ACSL4 (mechanism), expression during intramuscular preadipocyte adipogenesis of pig cells (same material and process) and siRNA knockdown effects of ACSL4 (loss of function). Article link (https://bmcgenomdata.biomedcentral.com/articles/10.1186/s12863-020-0836-7
). Authors could rewrite the novelty and background investigations of ACSL4 in intramuscular adipocytes.
Some other suggestions
Line 21- Acronym (comes after expanded form) should be in parenthesis, not the expanded form.
Line 26- Rewrite ‘our data showed that ACSL4 is a positive regulator of adipogenesis in intramuscular fat cells isolated from pigs’.
The style of the references should be followed as per the MDPI requirements.
Author Response
Response to Reviewer 3 Comments
Point 1: Throughout the manuscript, the authors mention the genetic polymorphism of ACSL4 and its impacts but have not performed any experiment in relation to it. I would suggest rewriting so that the message gets through clearly according to the results of the manuscript.
Response 1: Thank you very much for your comment. We revised the Introduction sections for the statement on ACSL4 gene polymorphisms. Please check the following text in the manuscript
“The variations of Acyl-CoA Synthetase Long-Chain Family Member 4 (ACSL4) gene locus are associated with intramuscular fat content in different pig populations, but the detailed molecular function of ACSL4 in pig intramuscular adipogenesis remain obscure.” (line 14-17)
“The genomic variations around the ACSL4 locus control the ACSL4 transcription but whether the fluctuations of ACSL4 expression levels could impact the intramuscular fat deposition remains unclear.” (line 47-50)
“Numerous studies indicated that the role of the ACSL4 gene is associated with the IMF content in different pig populations but the intrinsic biological function of ACSL4 in porcine intramuscular adipogenesis is still unclear.” (line 291-293)
Point 2: Figure 2A Have authors measured ACSL4 expression in intramuscular adipose tissue in addition to muscle and adipose? The authors explained liver-related high expression but did not comment on lungs (highest expression).
Response 2: Thank you very much for your comment. Regrettably, we only examined the expression levels of ACSL4 in muscle and adipose tissue. As for the high expression profile of ACSL4 in lung tissue, several studies have revealed that ACSL4 is mainly expressed in peroxisomes and the ER, while the process of lung lipid metabolism in lung tissue requires the involvement of substantial numbers of peroxisomes[1, 2]. We think that due to the critical effect of peroxisomes in lung tissue, this causes a hyper-expression profile of ACSL4 in the lung tissue. Please check the following text in the manuscript
“We also note that the high expression profile of ACSL4 in lung tissue. Several studies have revealed that ACSL4 is mainly expressed in peroxisomes and the ER, while the process of lung lipid metabolism in lung tissue requires the involvement of substantial numbers of peroxisomes. We think that due to the critical effect of peroxisomes in lung tissue, this causes a hyper-expression profile of ACSL4 in the lung tissue.” (line 308-313)
Reference:
- Grevengoed, T.J.,E.L. Klett;R.A. Coleman. Acyl-CoA metabolism and partitioning. Annu Rev Nutr. 2014, 34, 1-30.
- Karnati, S.;E. Baumgart-Vogt. Peroxisomes in mouse and human lung: their involvement in pulmonary lipid metabolism. Histochem Cell Biol. 2008, 130, 719-40.
Point 3: In Figure 1, the authors show a gradual increase in lipid accumulation from days 3 to 9 and claim that ACSL4 is positively associatedwith adipogenesis. However, ACSL4 expression increased during early adipogenesis and decreases at the maturation phase. The authors have not described or explained this difference.
Response 3: Thank you very much for your comment. The results of our tissue expression profile indicate that ACSL4 expression is low in mature adipose tissue, and we therefore propose that ACSL4 is dispensable for maintaining adipocyte and lipid deposition. We revised some words in the manuscript. Please check the following text in the manuscript.
“The loss-of-function experiments by siRNA knockdown in preadipocytes showed that ACSL4 is necessary for the full proceeding of the early adipogenic differentiation gene program.” (line 297-299)
“Our data also showed that ACSL4 expression remains low in the matured adipocytes, indicating that the ACSL4 is dispensable for the maintenance of fat cells and lipid storage.” (line 303-305)
Point 4: In the ACSL4 knockdown experiment results, authors have mentioned only downregulated DEGs. They could mention upregulated DEGs that may reveal genes suppressing adipogenesis.
Response 4: Thank you very much for your comment. We checked the upregulated DEGs according to your suggestion and indeed identified some suppressor genes for adipogenesis. The content of upregulated DEGs was added into the manuscript.
“Among the DEGs we identified many adipogenesis related genes were downregulated including MOGAT1 [18], IL10 [19], TFAP2B [20] and ACSL6 [21], while several negative regulators of adipogenesis such as FABP4[22] and SST [23] were upregulated.” (line 236-239)
Point 5: In general, the image quality should be increased. The authors could perform Oil Red O image analyses to measure the differences.
Response 5: Thank you for your suggestion. We added the oil red staining quantification in Figure1. Please check the Figure 1 in the manuscript
“F: The quantitation of oil red O (ORO) positive region for porcine intramuscular preadipocytes at 3, 6 and 9 days. Data are expressed as means+SEM or SD. Statistical significance was set at p < 0.05. ns indicated no significant difference * indicated p < 0.05, ** indicated p < 0.01, *** indicated p < 0.001.” (line 199-202)
Point 6: In Table 1, it is unclear why there is a break. Additionally, adding significance values could increase the validity of the differences and ease their identification. Moreover, a huge difference in Oleic Acid (most abundant FA in intramuscular fat) levels between overexpression and control samples is evident, authors haven’t discussed it.
Response 6: We acknowledge your suggestion and made corrections to the table form.
For the fatty acid composition analysis, we performed one thorough quantification of the lipid profiles of overexpression and the control samples. Unfortunately, we couldn’t provide the significance effect values based on the current data. Therefore, in the manuscript text we avoided to mention the significance of changes between the two treatments. We only described the trend and extend of the increase or decrease in some of the specific fatty acids based on the absolute values of the lipid species. In the future, we will assay the fatty acids composition in repeated samples from the same treatments to further define the ACSL4 role in the enhancement of meat value. We also revised the description of this result. We hope the reviewer could accept this explanation. Please check the following text in our manuscript.
“Quantification of the lipid classes revealed dramatic ACSL4-induced changes in the abundance of lipid species in several of the analyzed lipid classes.” (line 269-270)
Regarding the issue that overexpression of ACSL4 presents dramatic differences in oleic acid, we found that ACSL4 is an essential contributor for triggering cellular ferroptosis [3, 4]. Meanwhile, the study has shown that oleic acid reduced the mortality of ferric death cells [5]. We conclude that differences in intracellular oleic acid respond antagonistically to ACSL4 overexpression. Please check the following text in the manuscript:
“Regarding the dramatic differences in oleic acid, we found that ACSL4 is an essential contributor for triggering cellular ferroptosis [36, 37]. Meanwhile, the previous study has shown that oleic acid reduced the mortality of ferric death cells [13]. We conclude that differences in intracellular oleic acid may respond antagonistically to ACSL4 overexpression.” (line 335-338)
Reference:
- Magtanong, L.,P.J. Ko,M. To. Exogenous Monounsaturated Fatty Acids Promote a Ferroptosis-Resistant Cell State. Cell Chem Biol. 2019, 26, 420-432 e9.
- Yuan, H.,X. Li,X. Zhang. Identification of ACSL4 as a biomarker and contributor of ferroptosis. Biochem Biophys Res Commun. 2016, 478, 1338-43.
- Doll, S.,B. Proneth,Y.Y. Tyurina. ACSL4 dictates ferroptosis sensitivity by shaping cellular lipid composition. Nat Chem Biol. 2017, 13, 91-98.
Point 7: Authors have performed RNA-Seq but I could not find if they have submitted the data
Response 7: Thank you for the reminder. We uploaded the RNA-seq data on NCBI and revised the data availability statement. Please check the following text in our manuscript
“Data Availability Statement: All the sequencing data are available through BioProject database under the accession BioProject ID PRJNA791163” (line 363-364)
Point 8: Authors claim that their study is novel as studies investigating the role of ACSL4, mechanism of action and loss/gain of function explorations were lacking. However, a study has previously been published demonstrating the miRNA regulation of ACSL4 (mechanism), expression during intramuscular preadipocyte adipogenesis of pig cells (same material and process) and siRNA knockdown effects of ACSL4 (loss of function). Article link (https://bmcgenomdata.biomedcentral.com/articles/10.1186/s12863-020-0836-7). Authors could rewrite the novelty and background investigations of ACSL4 in intramuscular adipocytes.
Response 8: Thank you for your suggestion. We modified the statements and words in the introduction and abstract sections of the manuscript. Please check the following text in the manuscript
“The variations of Acyl-CoA Synthetase Long-Chain Family Member 4 (ACSL4) gene locus are associated with intramuscular fat content in different pig populations, but the detailed molecular function of ACSL4 in pig intramuscular adipogenesis remain obscure.” (line 14-17)
“Several previous GWAS studies identified Acyl-CoA Synthetase Long Chain Family Member 4 (ACSL4) gene as the candidate gene to regulate intramuscular fat content in different pig populations, but the underlying molecular function of ACSL4 in adipogenesis within pig skele-tal muscle is not fully investigated.” (line 21-24)
“A previous study reported that ACSL4 was involved in preadipocyte differentiation in pigs [9].” (line 49-50)
Point 9: Line 21- Acronym (comes after expanded form) should be in parenthesis, not the expanded form.
Response 9: Thank you for your suggestion. We revised this sentence. Please check the following text in our manuscript
“Several previous GWAS studies identified Acyl-CoA Synthetase Long Chain Family Member 4 (ACSL4) gene as the candidate gene to regulate intramuscular fat content in different pig populations, but the underlying molecular function of ACSL4 in adipogenesis within pig skele-tal muscle is not fully investigated.” (line 14-17)
Point 10: Line 26- Rewrite ‘our data showed that ACSL4 is a positive regulator of adipogenesis in intramuscular fat cells isolated from pigs’.
Response 10: Thank you for your suggestion. We rewrote this sentence in our paper. Please find the following text passages added to the manuscript:
“our data showed that ACSL4 is a positive regulator of adipogenesis in intramuscular fat cells isolated from pigs.” (line 26-27)
Point 11: The style of the references should be followed as per the MDPI requirements.
Response 11: Thank you for your suggestion. We revised the reference style in our manuscript. Please check the reference in our manuscript.

Reviewer 4 Report
Grammars should be checked thoroughly and corrected. Here are examples in the Abstract and Simple Summary
- The title, in pigs..
Ex. Line 16-17, is unknown.
- line 17, and provides…
- line 24, is still unknown.
- line 27, in pigs.
resources and aimed..
- The reference clones (access number in Genebank) of primers in qRT-PCR as well as for siRNA should be noted. Also, the amplicon length and optimized concentrations of primers should be noted.
- Where was the antibody against swine ACSL4 from?
- In Table 1, why fatty acid composition was expressed as μg/mL. It should be expressed as % based by relative weight or molar ratio. Even expressed as μg/mL in an absolute unit, mL is a volume unit not weight. It should be expressed as fatty acid weight or mole per mg or mg protein, or per mg or mg DNA.
- Where is the quantification result for lipid deposition?
- Where are the statistic comparisons in Table to conclude the significant effect?
- What is the n=? in each study for statistical comparison?
8. The authors need to further examine the differentiation program of adipogenesis rather than only focusing on the mechanism of lipid synthesis and deposition, which is a very late signature of adipogenesis.
Author Response
Point 1: The title, in pigs.
Point 2: Ex. Line 16-17, is unknown.
Point 3: line 17, and provides…
Point 4: line 24, is still unknown.
Response 1, 2, 3, 4: Thank you very much for these comments. Based on these above suggestions, we revised the manuscript in the corresponding places.
“ACSL4 directs intramuscular adipogenesis and fatty acid composition in pigs” (Title)
“The variations of Acyl-CoA Synthetase Long-Chain Family Member 4 (ACSL4) gene locus are associated with intramuscular fat content in different pig populations, but the detailed molecular function of ACSL4 in pig intramuscular adipogenesis remain obscure.” (line 14-17)
“Our study reveals the function of ACSL4 in pig intramuscular adipogenesis and provides new clues for improving the palatability of meat and enhancing the nutritional value of pork for human health.” (line 17-19)
“Several previous GWAS studies identified Acyl-CoA Synthetase Long Chain Family Member 4 (ACSL4) gene as the candidate gene to regulate intramuscular fat content in different pig populations, but the underlying molecular function of ACSL4 in adipogenesis within pig skele-tal muscle is not fully investigated.” (line 21-24)
Point 5: line 27, in pigs.
Response 5: Thank you very much for the comment. We think the preposition "from" is more appropriate in this position. Please check the following text in our manuscript
“our data showed that ACSL4 is a positive regulator of adipogenesis in intramuscular fat cells isolated from pigs” (line 26-27)
Point 6: The reference clones (access number in Genebank) of primers in qRT-PCR as well as for siRNA should be noted. Also, the amplicon length and optimized concentrations of primers should be noted.
Response 6: Thank you very much for this comment. We optimized the qPCR and siRNA primers, for example we added specific information to the method and a primer table. Please check the following text in our manuscript and Table S1
“Total RNA from porcine tissues and cells was isolated using total RNA extraction kit and Reverse Transcription kit (TIANGEN, China) according to manufacturer’s instructions. The qPCR primer of ACSL4, FASN, ACACβ and C/EBPα were designed according their cDNA sequence (GenBank NM_001033600.1; NM_007988.3; XM_006530111.4; NM_001287514.1) with the free online designtools (https://www.ncbi.nlm.nih.gov/tools/primer-blast/index.cgi). The detail of realtime-PCR primer sequences was in TableS1 ” (line 129-137).”
“In our study, the ACSL4 siRNA were designed according to its mRNA sequence (GenBank NM001038694.1) with the free online design tools (https://www.thermofiser.com/us/en/home/brands/invitrogen/ambion.html), and all the siRNA sequences were synthesized by Shanghai Genepharm Co., LTD. The sequences are as follows: ACSL4-siRNA: 5’-GTCCAAGAGATGAATTATATT-3’; a scramble sequence 5’-GTTCTCCGAACGTGTCACGT-3’ was also synthesized as a negative control. Transfection was conducted with Lipofectamine 3000 (Invitrogen, USA) according to manufacturer’s protocol.” (line 129-137)
Table S1 Primers used for real-time quantitative PCR
|
Primer name |
Sequence (5′→3′) |
Length(nt) |
production length |
GC (%) |
Tm |
|
ACSL4-F |
GATTGACAGAATCGTGTGGCG |
21 |
187 |
52.38 |
59.94 |
|
ACSL4-R |
CCCATGGAGATATTCTGTCCACC |
22 |
52.17 |
60.24 |
|
|
FASN-F |
TCACCTACGAGGCCATTGTG |
20 |
327 |
55 |
59.75 |
|
FASN-R |
GCTTCAGCAGGACGTTG |
17 |
58.82 |
55.9 |
|
|
ACACβ-F |
CAGAACCTGCATGACGGAGT |
20 |
155 |
55 |
60.04 |
|
ACACβ-R |
TGAGGACAAACATGGTCGGG |
20 |
55 |
55.96 |
|
|
C/EBPα-F |
GGCAAAGCCAAGAAGTCGGTA |
21 |
150 |
52.38 |
60.88 |
|
C/EBPα-R |
ATTGTCACTGGTCAGCTCCA |
20 |
50 |
52.94 |
Point 7: Where was the antibody against swine ACSL4 from?
Response 7: Thank you very much for this comment. In our study, a mouse anti-ACSL4 antibody was used as the primary antibody against swine ACSL4. The primary antibodies used for Western blotting were rabbit monoclonal antibody against the mouse ACSL4/FACL4 (ab205199, Abcam; dilution 1:1,000) due to the lack of specific antibodies against pig gene. We updated the content of the methods section. Please check the following text in our manuscript
“The primary antibodies used for Western blotting were rabbit monoclonal antibody against the mouse ACSL4/FACL4 (ab205199, Abcam; dilution 1:1,000), mouse monoclonal antibody against the human GAPDH (60004-1-Ig, Proteintech, Wuhan, China; dilution 1:2,000). HRP-conjugated secondary antibodies were anti-mouse and anti-rabbit IgG (7076 and 7074, respectively, Cell Signaling Technology; dilution 1:2,000). The intensity of protein bands was measured by the Image Lab software package.” (line 155-160)
Point 8: In Table 1, why fatty acid composition was expressed as μg/mL. It should be expressed as % based by relative weight or molar ratio. Even expressed as μg/mL in an absolute unit, mL is a volume unit not weight. It should be expressed as fatty acid weight or mole per mg or mg protein, or per mg or mg DNA.
Response 8: We really appreciate the reviewer’s valuable advice. To evaluate the fatty acid composition in cells, samples containing 1x107cells were harvested and resuspended in 1 mL of methyl alcohol, and all the liquids were further analyzed by gas chromatography spectrometry. For this reason, the fatty acid composition was expressed as μg/mL in our manuscript, this statement is not very proper. According to reviewer’s advice, the absolute amount of fatty acid composition was expressed as μg protein, and the number of cells was noted below Table 1, and we hope the reviewer could accept this explanation.
Point 9: Where is the quantification result for lipid deposition?
Response 9: Thank you for your suggestion. We added the oil red staining quantification in Figure1 and the content of method section. Please check the Figure 1 in the manuscript
“F: the quantitation of oil red O (ORO) positive region for porcine intramuscular preadipocytes at 3, 6 and 9 days Data are expressed as means+SEM or SD. Statistical significance was set at p < 0.05. ns indicated no significant difference * indicated p < 0.05, ** indicated p < 0.01, *** indicated p < 0.001.” (line 199-202)
“Oil Red O quantification was performed by measuring the positive area in three different fields (stained pixels were measured using Image J).” (line 102-103)
Point 10: Where are the statistic comparisons in Table to conclude the significant effect?
Response 10: Thank you for your comment. For the fatty acid composition analysis, we performed one thorough quantification of the lipid profiles of overexpression and the control samples. Unfortunately, we couldn’t provide the significance effect values based on the current data. Therefore, in the manuscript text we avoided to mention the significance of changes between the two treatments. We only described the trend and extend of the increase or decrease in some of the specific fatty acids based on the absolute values of the lipid species. In the future, we will assay the fatty acids composition in repeated samples from the same treatments to further define the ACSL4 role in the enhancement of meat value. We also revised the description of this result. We hope the reviewer could accept this explanation. Please check the following text in our manuscript.
“Quantification of the lipid classes revealed dramatic ACSL4-induced changes in the abundance of lipid species in several of the analyzed lipid classes.” (line 269-270)
Point 11: What is the n=? in each study for statistical comparison?
Response 11: Thank you for your comment. For statistical comparison, the qPCR and interference experiments were used with more than 3 sample replicates. Please check the following text in our manuscript.
“The numbers of biological replicates and technical repeats in each experimental group were three or more. Statistical analyses were performed using Graphpad Prism (Graphpad Software). Data are expressed as means+SEM or SD. P-value of 0.05 is considered statistically significant.” (line 176-179)
Point 12: The authors need to further examine the differentiation program of adipogenesis rather than only focusing on the mechanism of lipid synthesis and deposition, which is a very late signature of adipogenesis.
Response 12: We really appreciate the reviewer’s valuable advice. Unfortunately, we think it is challenge to investigate the mechanism for differentiation of adipogenesis using pigs as experimental animals. In the subsequent research project, we would consider the research on the differentiation mechanism of adipogenesis in mice. In addition, in terms of the results of lipid synthesis and deposition, our intention is to unravel the role of ACSL4 in the intramuscular lipogenesis of pigs. More specifically, the investigation on the palatability and improvement of the nutritional value of pork. Lipid deposition and fatty acid (FA) composition in muscle are determinants of meat quality, particularly regarding flavour, juiciness and tenderness [1-3]. Therefore, the purpose of our ACSL4 overexpression experiment was to emphasize the importance of ACLS4 for meat fatty acids and thus provide the insight of improving meat flavour. We hope the reviewer could accept this explanation.
- Wood, J.D.,M. Enser,A.V. Fisher. Fat deposition, fatty acid composition and meat quality: A review. Meat Sci. 2008, 78, 343-58.
- Burnett, D.D.,J.F. Legako,K.J. Phelps. Biology, strategies, and fresh meat consequences of manipulating the fatty acid composition of meat. J Anim Sci. 2020, 98.
- Wu, G.,X. Shi,J. Zhou. Differential expression of meat quality and intramuscular fat deposition related genes in Hanjiang black pigs. Acta Biochim Biophys Sin (Shanghai). 2015, 47, 145.

Round 2
Reviewer 1 Report
Please remove all non-English characters.
Author Response
Review Report (Reviewer 1)
Point1:Please remove all non-English characters.
Response1: Thank you very much for this comment. We removed all non-English characters,please check our manuscript.

Reviewer 3 Report
Dear Authors,
Thank you modifying the manuscript and providing the additional needed information. All the comments have been responded properly.
The RNA-Seq data submission is not available on NCBI. Authors should make sure the data is submitted/available before the further processing.
Author Response
Review Report (Reviewer 3)
Point1: Thank you modifying the manuscript and providing the additional needed information. All the comments have been responded properly.
The RNA-Seq data submission is not available on NCBI. Authors should make sure the data is submitted/available before the further processing.
Response1: Thank you very much for this comment. We submitted the data on Bioproject, but the release date of data is December 2024. Please review the content of Email from NCBI.

Reviewer 4 Report
- In Table 1, if the results were expressed as absolute unit, the title should be changed to The fatty acid composition of samples. According to the description in the Materials methods, fatty acid analysis was peroformed by GC method.So, what is the lipid extraction method? by Chloroform/methanol? If so, the fatty acid results should be noted as total lipids and the results should be noted as means ± SD (or SE) with statistical significance.
Author Response
Review Report (Reviewer 4)
Point 1: In Table 1, if the results were expressed as absolute unit, the title should be changed to the fatty acid composition of samples. According to the description in the Materials methods, fatty acid analysis was performed by GC method. So, what is the lipid extraction method? by Chloroform/methanol? If so, the fatty acid results should be noted as total lipids and the results should be noted as means ± SD (or SE) with statistical significance.
Response 1: Thank you very much for this comment. We used chloroform-methanol (2:1) solution to extract lipid from sample and we also added more detail in the method section. Please check the following text in method section of manuscript.
“Adipose precursor cells were transfected with adenovirus vector. 1 x 107 cells were harvested for each cell sample and resuspended with 1 ml of chloroform-methanol (2:1) solution. The samples were snap frozen in liquid nitrogen and then dissolved at room temperature. The samples were grinded at 60 Hz for 2 min, followed by ultrasonic treatment for 30 min. The samples were centrifuged at 13,000 rpm for 5 min at 4 °C and the supernatant was collected and mixed with 2 mL of 1% methanolic sulphate. The mixed samples were esterified at 80 °C for 30 minutes and 1 mL of hexane was added for extraction. The samples were washed at 4 °C with 5 mL dd H2O, followed by sodium sulphate powder anhydrous to remove extra water. The supernatant was collected by centrifuging at 13,000 rpm for 5 min and mixed with 25 μL of methyl salicylate. The supernatant from mixture was placed on a DB5 capillary column (30 m x 0.25 mm) for gas chromatography mass spectrometry (Agilent 6890N/5975B) analysis.” (Line 170-181)
For fatty acid results, we collected the same cell number samples for gas chromatography by cell counting and added unit descriptions in the note of the table 1. Please check the table 1 in manuscript.
“Note: cell sample is 1x107 cells and was resuspended in 1 mL of chloroform-methanol solution. Units in the table means μg /1x107 cells.” (Line 296-297)
For the fatty acid composition analysis, we performed one thorough quantification of the lipid profiles of overexpression and the control samples. Unfortunately, we couldn’t provide the statistical significance values based on the current data. Therefore, in the revised manuscript main text we avoided to mention the significance of changes between the treated and control. We toned down the wording of expression to only described the trend and extend of the increase or decrease in several of the specific fatty acids based on the absolute values of the lipid species. In the future, we will assay the fatty acids composition in repeated samples from the same treatments to further define the ACSL4 role in the enhancement of meat value.
